# Park: An Open Platform for Learning-Augmented Computer Systems

**Hongzi Mao, Parimarjan Negi, Akshay Narayan, Hanrui Wang, Jiacheng Yang,
Haonan Wang, Ryan Marcus, Ravichandra Addanki, Mehrdad Khani, Songtao He,
Vikram Nathan, Frank Cangialosi, Shaileshh Bojja Venkatakrishnan,
Wei-Hung Weng, Song Han, Tim Kraska, Mohammad Alizadeh**
MIT Computer Science and Artificial Intelligence Laboratory
`park-project@csail.mit.edu`

## Abstract

We present Park, a platform for researchers to experiment with Reinforcement
Learning (RL) for computer systems. Using RL for improving the performance
of systems has a lot of potential, but is also in many ways very different from,
for example, using RL for games. Thus, in this work we first discuss the unique
challenges RL for systems has, and then propose Park an open extensible platform,
which makes it easier for ML researchers to work on systems problems. Currently,
Park consists of 12 real world system-centric optimization problems with one
common easy to use interface. Finally, we present the performance of existing RL
approaches over those 12 problems and outline potential areas of future work.

## 1 Introduction

Deep reinforcement learning (RL) has emerged as a general and powerful approach to sequential
decision making problems in recent years. However, real-world applications of deep RL have thus far
been limited. The successes, while impressive, have largely been confined to controlled environments,
such as complex games [70, 78, 91, 97, 100] or simulated robotics tasks [45, 79, 84]. This paper
concerns applications of RL in computer systems, a relatively unexplored domain where RL could
provide significant real-world benefits.

Computer systems are full of sequential decision-making tasks that can naturally be expressed
as Markov decision processes (MDP). Examples include caching (operating systems), congestion
control (networking), query optimization (databases), scheduling (distributed systems), and more
(§2). Since real-world systems are difficult to model accurately, state-of-the-art systems often rely on
human-engineered heuristic algorithms that can leave significant room for improvement [69]. Further,
these algorithms can be complex (e.g., a commercial database query optimizer involves hundreds of
rules [14]), and are often difficult to adapt across different systems and operating environments [63,
66] (e.g., different workloads, different distribution of data in a database, etc.). Furthermore, unlike
control applications in physical systems, most computer systems run in software on readily-available
commodity machines. Hence the cost of experimentation is much lower than physical environments
such as robotics, making it relatively easy to generate abundant data to explore and train RL models.
This mitigates (but does not eliminate) one of the drawbacks of RL approaches in practice — their
high sample complexity [7]. The easy access to training data and the large potential benefits have
attracted a surge of recent interest in the systems community to develop and apply RL tools to various
problems [17, 24, 32, 34, 48, 51, 54, 61–63, 66, 68, 69].

From a machine learning perspective, computer systems present many challenging problems for RL.
The landscape of decision-making problems in systems is vast, ranging from centralized control
problems (e.g., a scheduling agent responsible for an entire computer cluster) to distributed multi-

agent problems where multiple entities with partial information collaborate to optimize system performance (e.g., network congestion control with multiple connections sharing bottleneck links). Further, the control tasks manifest at a variety of timescales, from fast, reactive control systems with sub-second response-time requirements (e.g., admission/eviction algorithms for caching objects in memory) to longer term planning problems that consider a wide range of signals to make decisions (e.g., VM allocation/placement in cloud computing). Importantly, computer systems give rise to new challenges for learning algorithms that are not common in other domains (§3). Examples of these challenges include time-varying state or action spaces (e.g., dynamically varying number of jobs and machines in a computer cluster), structured data sources (e.g., graphs to represent data flow of jobs or a network's topology), and highly stochastic environments (e.g., random time-varying workloads). These challenges present new opportunities for designing RL algorithms. For example, motivated by applications in networking and queuing systems, recent work [64] developed new general-purpose control variates for reducing variance of policy gradient algorithms in "input-driven" environments, in which the system dynamics are affected by an exogenous, stochastic process.

Despite these opportunities, there is relatively little work in the machine learning community on algorithms and applications of RL in computer systems. We believe a primary reason is the lack of good benchmarks for evaluating solutions, and the absence of an easy-to-use platform for experimenting with RL algorithms in systems. Conducing research on learning-based systems currently requires significant expertise to implement solutions in real systems, collect suitable real-world traces, and evaluate solutions rigorously. The primary goal of this paper is to lower the barrier of entry for machine learning researchers to innovate in computer systems.

We present Park, an open, extensible platform that presents a common RL interface to connect to a suite of 12 computer system environments (§4). These representative environments span a wide variety of problems across networking, databases, and distributed systems, and range from centralized planning problems to distributed fast reactive control tasks. In the backend, the environments are powered by both real systems (in 7 environments) and high fidelity simulators (in 5 environments). For each environment, Park defines the MDP formulation, e.g., events that triggers an MDP step, the state and action spaces and the reward function. This allows researchers to focus on the core algorithmic and learning challenges, without having to deal with low-level system implementation issues. At the same time, Park makes it easy to compare different proposed learning agents on a common benchmark, similar to how OpenAI Gym [19] has standardized RL benchmarks for robotics control tasks. Finally, Park defines a RPC interface [92] between the RL agent and the backend system, making it easy to extend to more environments in the future.

We benchmark the 12 systems in Park with both RL methods and existing heuristic baselines (§5). The experiments benchmark the training efficiency and the eventual performance of RL approaches on each task. The empirical results are mixed: RL is able to outperform state-of-the-art baselines in several environments where researchers have developed problem-specific learning methods; for many other systems, RL has yet to consistently achieve robust performance. We open-source Park as well as the RL agents and baselines in https://github.com/park-project/park.

## 2    Sequential Decision Making Problems in Computer Systems

Sequential decision making problems manifest in a variety of ways across computer systems disciplines. These problems span a multi-dimensional space from centralized vs. multi-agent control to reactive, fast control loops vs. long-term planning. In this section, we overview a sample of problems from each discipline and how to formulate them as MDPs. Appendix A provides further examples and a more formal description of the MDPs that we have implemented in Park.

**Networking.** Computer network problems are fundamentally distributed, since they interconnect independent users. One example is congestion control, where hosts in the network must each determine the rate to send traffic, accounting for both the capacity of the underlying network infrastructure and the demands of other users of the network. Each network connection has an agent (typically at the sender side) setting the sending rate based on how previous packets were acknowledged. This component is crucial for maintaining a large throughput and low delay.

Another example at the application layer is bitrate adaptation in video streaming. When streaming videos from content provider, each video is divided into multiple chunks. At watch time, an agent decides the bitrate (affecting resolution) of each chunk of the video based on the network (e.g.,

bandwidth and latency measurements) and video characteristics (e.g., type of video, encoding scheme, etc.). The goal is to learn a policy that maximizes the resolution while minimizing chance of stalls (when slow network cannot download a chunk fast enough).

**Databases.** Databases seek to efficiently organize and retrieve data in response to user requests. To efficiently organize data, it is important to index, or arrange, the data to suit the retrieval patterns. An indexing agent could observe query patterns and accordingly decide how to best structure, store, and over time, re-organize the data.

Another example is query optimization. Modern query optimizers are complex heuristics which use a combination of rules, handcrafted cost models, data statistics, and dynamic programming, with the goal to re-order the query operators (e.g., joins, predicates) to ultimately lower the execution time. Unfortunately, existing query optimizers do not improve over time and do not learn from mistakes. Thus, they are an obvious candidate to be optimized through RL [66]. Here, the goal is to learn a query optimization policy based on the feedback from optimizing and running a query plan.

**Distributed systems.** Distributed systems handle computations that are too large to fit on one computer; for example, the Spark framework for big-data processing computes results across data stored on multiple computers [107]. To efficiently perform such computations, a job scheduler decides how the system should assign compute and memory resources to jobs to achieve fast completion times. Data processing jobs often have complex structure (e.g., Spark jobs are structured as dataflow graphs, Tensorflow models are computation graphs). The agent in this case observes a set of jobs and the status of the compute resources (e.g., how each job is currently assigned). The action decides how to place jobs onto compute resources. The goal is to complete the jobs as soon as possible.

**Operating systems.** Operating systems seek to efficiently multiplex hardware resources (compute, memory, storage) amongst various application processes. One example is providing a memory hierarchy: computer systems have a limited amount of fast memory and relatively large amounts of slow storage. Operating systems provide caching mechanisms which multiplex limited memory amongst applications which achieve performance benefits from residency in faster portions of the cache hierarchy. In this setting, an RL agent can observe the information of both the existing objects in the cache and the incoming object; it then decides whether to admit the incoming object and which stale objects to evict from the cache. The goal is to maximize the cache hit rate (so that more application reads occur from fast memory) based on the access pattern of the objects.

Another example is CPU power state management. Operating systems control whether the CPU should run at an increased clock speed and boost application performance, or save energy with at a lower clock speed. An RL agent can dynamically control the clock speed based on the observation of how each application is running (e.g., is an application CPU bound or network bound, is the application performing IO tasks). The goal is to maintain high application performance while reducing the power consumption.

# 3 RL for Systems Characteristics and Challenges

In this section, we explain the unique characteristics and challenges that often prevent off-the-shelf RL methods from achieving strong performance in different computer system problems. Admittedly, each system has its own complexity and contains special challenges. Here, we primarily focus on the common challenges that arise across many systems in different stages of the RL design pipeline.

## 3.1 State-action Space

**The needle-in-the-haystack problem.** In some computer systems, the majority of the state-action space presents little difference in reward feedback for exploration. This provides no meaningful gradient during RL training, especially in the beginning, when policies are randomly initialized. Network congestion control is a classic example: even in the simple case of a fixed-rate link, setting the sending rate above the available network bandwidth saturates the link and the network queue. Then, changes in the sending rate above this threshold result in an equivalently bad throughput and delay, leading to constant, low rewards. To exit this bad state, the agent must set a low sending rate for multiple *consecutive* steps to drain the queue before receiving any positive reward. Random exploration is not effective at learning this behavior because any random action can easily overshadow several good actions, making it difficult to distinguish good action sequences from bad ones. Circuit

|  | GCN direct | GCN transfer | LSTM direct | LSTM transfer | Random |
|---|---|---|---|---|---|
| CIFAR-10 [52] | **1.73** $\pm$ 0.41 | **1.81** $\pm$ 0.39 | **1.78** $\pm$ 0.38 | 1.97 $\pm$ 0.37 | 2.15 $\pm$ 0.39 |
| Penn Tree Bank [65] | **4.84** $\pm$ 0.64 | **4.96** $\pm$ 0.63 | 5.09 $\pm$ 0.63 | 5.28 $\pm$ 0.6 | 5.42 $\pm$ 0.57 |
| NMT [11] | **1.98** $\pm$ 0.55 | **2.07** $\pm$ 0.51 | 2.16 $\pm$ 0.56 | 2.88 $\pm$ 0.66 | 2.47 $\pm$ 0.48 |

**Table 1:** Generalizability of GCN and LSTM state representation in the Tensorflow device placement environment. The numbers are average runtime in seconds. $\pm$ spans one standard deviation. Bold font indicate the runtime is within 5% of the best runtime. "Transfer" means testing on unseen models in the dataset.

design is another example: when *any* of the circuit components falls outside the operating region (the exact boundary is unknown before invoking the circuit simulator), the circuit cannot function properly and the environment returns a constant bad reward. As a result, exploring these areas provides little gradient for policy training.

In these environments, using domain-knowledge to confine the search space helps to train a strong policy. For example, we observed significant performance improvements for network congestion control problems when restricting the policy (see also Figure 4d). Also, environment-specific reward shaping [76] or bootstrapping from existing policies [41, 90] can improve policy search efficiency.

**Representation of state-action space.** When designing RL methods for problems with complex structure, properly encoding the state-action space is the key challenge. In some systems, the action space grows exponentially large as the problem size increases. For example, in switch scheduling, the action is a bijection mapping (a matching) between input and output ports — a standard 32-port would have 32! possible matching. Encoding such a large action space is challenging and makes it hard to use off-the-shelf RL agents. In other cases, the size of the action space is constantly changing over time. For example, a typical problem is to map jobs to machines. In this case, the number of possible mappings and thus, actions increases with the number of new jobs in the system.

Unsurprisingly, domain specific representations that capture inherent structure in the state space can significantly improve training efficiency and generalization. For example, Spark jobs, Tensorflow components, and circuit design are to some degree dataflow graphs. For these environments, leveraging Graph Convolutional Neural Networks (GCNs) [50] rather than LSTMs can significantly improves generalization (see Table 1). However, finding the right representation for each problem is a central challenge, and for some domains, e.g., query optimization, remains largely unsolved.

## 3.2 Decision Process

**Stochasticity in MDP causing huge variance.** Queuing systems environments (e.g., job scheduling, load balancing, cache admission) have dynamics partially dictated by an exogenous, stochastic *input process*. Specifically, their dynamics are governed not only by the decisions made within the system, but also the arrival process that brings work (e.g., jobs, packets) into the system. In these environments, the stochasticity in the input process causes huge variance in the reward.

For illustration, consider the load balancing example in Figure 1. If the arrival sequence after time $t$ consists of a burst of large jobs (e.g., job sequence 1), the job queue will grow and the agent will receive low rewards. In contrast, a stream of lightweight jobs

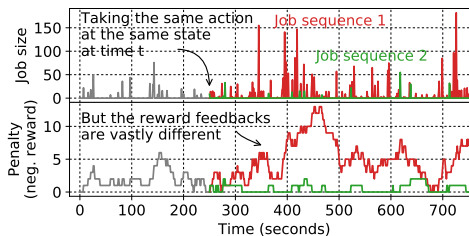

**Figure 1:** Illustrative example of load balancing showing how different instances of a stochastic input process can have vastly different rewards. After time $t$, we sample two job arrival sequences from a Poisson process. Figure adopted from [63].

(e.g., job sequence 2) will lead to short queues and large rewards. The problem is that this difference in reward is independent of the action at time $t$; rather, it is caused purely by the randomness in the job arrival process. In these environments, the agents cannot tell whether two reward feedbacks differ due to disparate input processes, or due to the quality of the actions. As a result, standard methods for estimating the value of an action suffer from high variance.

Prior work proposed an input-dependent baseline that effectively reduces the variance from the input process [64]. Figure 5 in [64] shows the policy improvement when using input-dependent

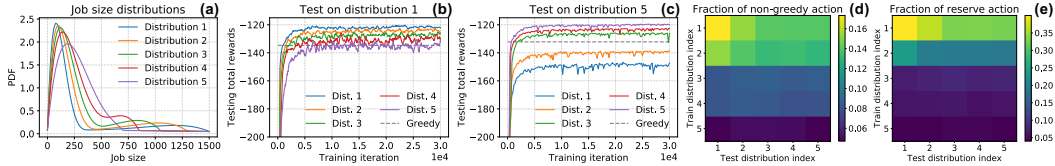

**Figure 2:** Demonstration of the gap between simulation and reality in the load balancing environment. (a) Distribution of job sizes in the training workload. (b, c) Testing agents on a particular distribution. An agent trained with distribution 5 is more robust than one trained with distribution 1. (d, e) A "reservation" policy that keeps a server empty for small jobs. Such a policy overfits distribution 1 and is not robust to workload changes.

baselines in the load-balancing and adaptive video streaming environments. However, the proposed training implementations ("multi-value network" and "meta baseline") are tailored for policy gradient methods and require the environments to have a repeatable input process (e.g., in simulation, or real systems with controllable input sequence). Thus, coping with input-driven variance remains an open problem for value-based RL methods and for environments with uncontrollable input processes.

**Infinite horizon problems.** In practice, production computer systems (e.g., Spark schedulers, load balancers, cache controllers, etc.) are long running and host services indefinitely. This creates an infinite horizon MDP [13] that prevents the RL agents from performing episodic training. In particular, this creates difficulties for bootstrapping a value estimation since there is no terminal state to easily assign a known target value. Moreover, the discounted total reward formulation in the episodic case might not be suitable — an action in a long running system can have impact beyond a fixed discounting window. For example, scheduling a large job on a slow server blocks future small jobs (affecting job runtime in the rewards), no matter whether the small jobs arrive immediately after the large job or much farther in the future over the course of the lifetime of the large job. Average reward RL formulations can be a viable alternative in this setting (see §10.3 in [93] for an example).

## 3.3 Simulation-Reality Gap

Unlike training RL in simulation, robustly deploying a trained RL agent or directly training RL on an actual running computer systems has several difficulties. First, discrepancies between simulation and reality prevent direct generalization. For example, in database query optimization, existing simulators or query planners use offline cost models to predict query execution time (as a proxy for the reward). However, the accuracy of the cost model quickly degrades as the query gets more complex due to both variance in the underlying data distribution and system-specific artifacts [53].

Second, interactions with some real systems can be slow. In adaptive video streaming, for example, the agent controls the bitrate for each chunk of a video. Thus, the system returns a reward to the agent only after a video chunk is downloaded, which typically takes a few seconds. Naively using the same training method from simulation (as in Figure 4a) would take a single-threaded agent more than 10 years to complete training in reality.

Finally, live training or directly deploying an agent from simulation can degrade the system performance. Figure 2 describes a concrete example for load balancing. The reason is that based on the bimodal distribution in the beginning, it learns to reserve a certain server for small jobs. However, when the distribution changes, blindly reserving a server wastes compute resource and reduces system throughput. Therefore, to deploy training algorithms online, these problems require RL to train robust policies that ensure safety [2, 33, 49].

## 3.4 Understandability over Existing Heuristics

As in other areas of ML, interpretability plays an important role in making learning techniques practical. However, in contrast to perception-based problems or games, for system problems, many reasonable good heuristics exist. For example, every introductory course to computer science features a basic scheduling algorithm such as FIFO. These heuristics are often easy to understand and to debug, whereas a learned approach is often not. Hence, making learning algorithms in systems as debuggable and interpretable as existing heuristics is a key challenge. Here, a unique opportunity is to build hybrid solutions, which combine learning-based techniques with traditional heuristics. Existing heuristics can not only help to bootstrap certain problems, but also help with safety and

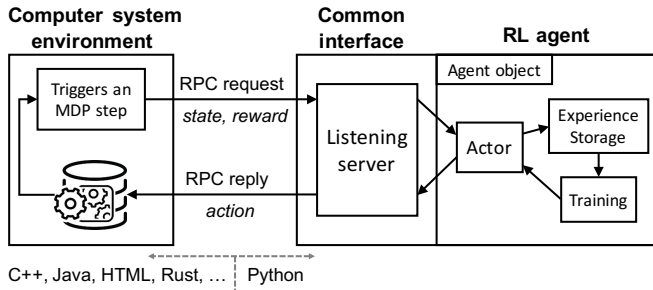

| Computer system environment | Common interface | RL agent |
|---|---|---|

**Algorithm 1** Interface for real system interaction.

```
1: def env.run(agent):
2:     while not done:
3:         state, reward, done = server.listen()
4:         # reward for the previous action
5:         action = agent.act(state, reward, done)
6:         server.reply(action)
```

**Algorithm 2** Interface for simulated interaction.

```
1: def env.step(action):
2:     # OpenAI Gym style of interaction
3:     server.reply(action)
4:     state, reward, done = server.listen()
5:     return state, reward, done
```

**Figure 3:** Park architects an RL-as-a-service design paradigm. The computer system connects to an RL agent through a canonical request/response interface, which hides the system complexity from the RL agent. Algorithm 1 describes a cycle of the system interaction with the RL agent. By wrapping with an agent-centric environment in Algorithm 2, Park's interface also supports OpenAI Gym [19] like interaction for simulated environments.

generalizability. For example, a learned scheduling algorithm could fall back to a simple heuristic if it detects that the input distribution significantly drifted.

## 4 The Park Platform

Park follows a standard request-response design pattern. The backend system runs continuously and periodically send requests to the learning agent to take control actions. To connect the systems to the RL agents, Park defines a common interface and hosts a server that listens for requests from the backend system. The backend system and the agent run on different processes (which can also run on different machines) and they communicate using remote procedure calls (RPCs). This design essentially structures RL as a service. Figure 3 provides an overview of Park.

**Real system interaction loop.** Each system defines its own events to trigger an MDP step. At each step, the system sends an RPC request that contains the current state and a reward corresponding to the *last* action. Upon receiving the request, the Park server invokes the RL agent. The implementation of the agent is up to the users (e.g., feature extraction, training process, inference methods). In Figure 3, Algorithm 1 depicts this interaction process. Notice that invoking the agent incurs a physical delay for the RPC response from the server. Depending on the underlying implementation, the system may or may not wait synchronously during this delay. For non-blocking RPCs, the state observed by the agent can be stale (which typically would not occur in simulation). On the other hand, if the system makes blocking RPC requests, then taking a long time to compute an action (e.g., while performing MCTS search [91]) can degrade the system performance. Designing high-performance RL training or inference agents in a real computer system should explicitly take this delay factor into account.

**Wrapper for simulated interaction.** By wrapping the request-response interface with a shim layer, Park also supports an "agent-centric" style of interaction advocated by OpenAI Gym [19]. In Figure 3, Algorithm 2 outlines this option in simulated system environments. The agent explicitly steps the environment forward by sending the action to the underlying system through the RPC response. The interface then waits on the RPC server for the next action request. With this interface, we can directly reuse existing off-the-shelf RL training implementations benchmarked on Gym [26].

**Scalability.** The common interface allows multiple instances of a system environment to run concurrently. These systems can generate the experience in parallel to speed up RL training. As a concrete example, to implement IMPALA [28] style of distributed RL training, the interface takes multiple actor instance at initialization. Each actor corresponds to an environment instance. When receiving an RPC request, the interface then uses the RPC request ID to route the request to the corresponding actor. The actor reports the experience to the learner (globally maintained for all agents) when the experience buffer reaches the batch size for training and parameter updating.

**Environments.** Table 2 provides an overview of 12 environments that we have implemented in Park. Appendix A contains the detailed descriptions of each problem, its MDP definition, and explanations of why RL could provide benefits in each environment. Seven of the environments use real systems in the backend (see Table 2). For the remaining five environments, which have well-understood dynamics, we provide a simulator to facilitate easier setup and faster RL training. For these simulated environments, Park uses real-world traces to ensure that they mimic their respective real-world environments faithfully. For example, for the CDN memory caching environment, we

| Environment | Type | State space | Action space | Reward | Step time | Challenges (§3) |
|---|---|---|---|---|---|---|
| Adaptive video streaming | Real/sim | Past network throughput measurements, playback buffer size, portion of unwatched video | Bitrate of the next video chunk | Combination of resolution and stall time | Real: ~3s Sim: ~1ms | Input-driven variance, slow interaction time |
| Spark cluster job scheduling | Real/sim | Cluster and job information as features attached to each node of the job DAGs | Node to schedule next | Runtime penalty of each job | Real: ~5s Sim: ~5ms | Input-driven variance, state representation, infinite horizon, reality gap |
| SQL database query optimization | Real | Query graph with predicate and table features on nodes, join attributes on edges | Edge to join next | Cost model or actual query time | ~5s | State representation, reality gap |
| Network congestion control | Real | Throughput, delay and packet loss | Congestion window and pacing rate | Combination of throughput and delay | ~10ms | Sparse space for exploration, safe exploration, infinite horizon |
| Network active queue management | Real | Past queuing delay, enqueue/dequeue rate | Drop rate | Combination of throughput and delay | ~50ms | Infinite horizon, reality gap |
| Tensorflow device placement | Real/sim | Current device placement and runtime costs as features attached to each node of the job DAGs | Updated placement of the current node | Penalty of runtime and invalid placement | Real: ~2s Sim: ~10ms | State representation, reality gap |
| Circuit design | Sim | Circuit graph with component ID, type and static parameters as features on the node | Transistor sizes, capacitance and resistance of each node | Combination of bandwidth, power and gain | ~2s | State representation, sparse space for exploration |
| CDN memory caching | Sim | Object size, time since last hit, cache occupancy | Admit/drop | Byte hits | ~2ms | Input-driven variance, infinite horizon, safe exploration |
| Multi-dim database indexing | Real | Query workload, stored data points | Layout for data organization | Query throughput | ~30s | State/action representation, infinite horizon |
| Account region assignment | Sim | Account language, region of request, set of linked websites | Account region assignment | Serving cost in the future | ~1ms | State/action representation |
| Server load balancing | Sim | Current load of the servers and the size of incoming job | Server ID to assign the job | Runtime penalty of each job | ~1ms | Input-driven variance, infinite horizon, safe exploration |
| Switch scheduling | Sim | Queue occupancy for input-output port pairs | Bijection mapping from input ports to output ports | Penalty of remaining packets in the queue | ~1ms | Action representation |

**Table 2:** Overview of the computer system environments supported by Park platform.

use an open dataset containing 500 million requests, collected from a public CDN serving top-ten US websites [15]. Given the request pattern, precisely simulating the dynamics of the cache (hits and evictions) is straightforward. Moreover, for each system environment, we also summarize the potential challenges from §3.

**Extensibility.** Adding a new system environment in Park is straightforward. For a new system, it only needs to specify (1) the state-action space definition (e.g., tensor, graph, powerset, etc.), (2) the event to trigger an MDP step, at which it sends an RPC request and (3) the function to calculate the reward feedback. From the agent's perspective, as long as the state-action space remains similar, it can use the same RL algorithm for the new environment. The common interface decouples the development of an RL agent from the complexity of the underlying system implementations.

## 5  Benchmark Experiments

We train the agents on the system environments in Park with several existing RL algorithms, including DQN [70], A2C [71], Policy Gradient [94] and DDPG [55]. When available, we also provide the existing heuristics and the optimal policy (specifically designed for each environment) for comparison. The details of hyperparameter tunings, agent architecture and system configurations are in Appendix B. Figure 4 shows the experiment results. As a sanity check, the performance of the RL policy improves over time from random initialization in all environments.

**Room for improvement.** We highlight system environments that exhibit unstable learning behaviors and potentially have large room for performance improvement. We believe that the instability observed in some of the environments are due to fundamental challenges that require new training procedure. For example, the policy in Figure 4h is unable to smoothly converge partially because of the variance caused by the cache arrival input sequence (§3.2). For database optimization in Figure 4c, RL methods that make one-shot decisions, such as DQN, do not converge to a stable policy; combining with explicit search [66] may improve the RL performance. In network congestion control, random exploration is inefficient to search the large state space that provides little reward gradient. This is because unstable control policies (which widely spans the policy space) cannot drain

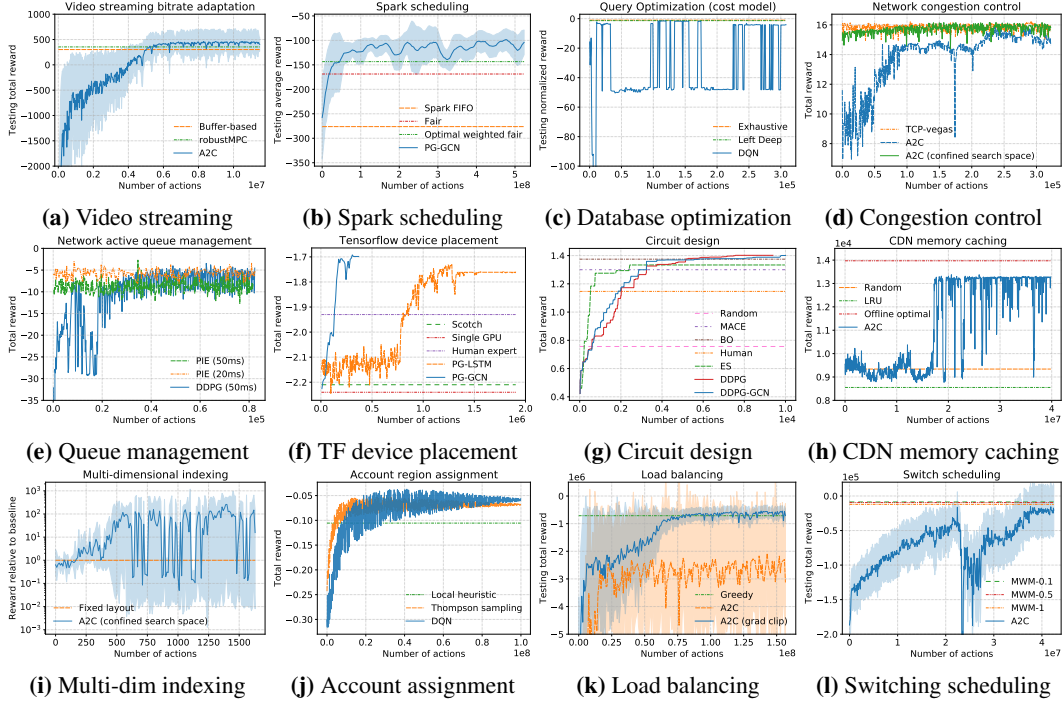

**Figure 4:** Benchmarks of the existing standard RL algorithms on Park environments. In y-axes, "testing" means the agents are tested with unseen settings in the environment (e.g., newly sampled workload unseen during training, unseen job patterns to schedule, etc.). The heuristic or optimal policies are provided as comparison.

the network queue fast enough and results in indistinguishable (e.g., delay matches max queuing delay) poor rewards (as discussed in §3.1). Confining the search space with domain knowledge significantly improves learning efficiency in Figure 4d (implementation details in Appendix B.2). For Tensorflow device placement in Figure 4f, using graph convolutional neural networks (GCNs) [50] for state encoding is natural to the problem setting and allows the RL agent to learn more than 5 times faster than using LSTM encodings [68]. Using more efficient encoding may improve the performance and generalizability further.

For some of the environments, we were forced to simplify the task to make it feasible to apply standard RL algorithms. Specifically, in CDN memory caching (Figure 4h), we only use a small 1MB cache (typical CDN caches are over a few GB); a large cache causes the reward (i.e., cache hit/miss) for an action to be significantly delayed (until the object is evicted from the cache, which can take hundreds of thousands of steps in large caches) [15]. For account region assignment in Figure 4j, we only allocate an account at initialization (without further migration). Active migration at runtime requires a novel action encoding (how to map any account to any region) that is scalable to arbitrary size of the action space (since the number of accounts keep growing). In Figure 4l, we only test with a small switch with $3 \times 3$ ports, because standard policy network cannot encode or efficiently search the exponentially large action space when the number of ports grow beyond $10 \times 10$ (as described in §3.1). These tasks are examples where applying RL in realistic settings may require inventing new learning techniques (§3).

## 6 Conclusion

Park provides a common interface to a wide spectrum of real-world systems problems, and is designed to be easily-extensible to new systems. Through Park, we identify several unique challenges that may fundamentally require new algorithmic development in RL. The platform makes systems problems easily-accessible to researchers from the machine learning community so that they can focus on the algorithmic aspect of these challenges. We have open-sourced Park along with the benchmark RL agents and the existing baselines in `https://github.com/park-project`.

**Acknowledgments.** We thank the anonymous NeurIPS reviewers for their constructive feedback. This work was funded in part by the NSF grants CNS-1751009, CNS-1617702, a Google Faculty Research Award, an AWS Machine Learning Research Award, a Cisco Research Center Award, an Alfred P. Sloan Research Fellowship, and sponsors of the MIT DSAIL lab.

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
