[Supplementary Material]

# Appendices

## A  Detailed descriptions of Park environments

We describe the details of each system environment in Park. Formulating the MDP is an important, problem-specific step for applying RL to systems. Our guiding principle is to provide the RL agent with all the information and actions available to existing baselines schemes in that environment, such that the agent could at least express existing human-engineered policies. In most cases, the MDP formulations are straightforward and self-explanatory. However, some are more subtle (e.g., the Spark scheduling and TF device placement), and in these cases we adopt the formulations from prior work. In the following, each description is structured to follow the problem background, MDP abstraction of the system interaction, the existing system-specific baseline heuristic approach, and how RL is suitable for the system problem.

**Adaptive video streaming.** The volume of video streaming has reached almost $60\%$ of all the Internet traffic [87]. Streaming video over variable-bandwidth networks (e.g., cellular network) requires the client to adapt the video bitrate to optimize the user experience. In industrial DASH standard [4], videos are divided into multiple chunks, each of which represents a few seconds of the overall video playback. Each chunk is encoded at several discrete bitrates, where a higher bitrate implies a higher resolution and thus a larger chunk size. For this problem, each MDP episode is a video playback with a particular network trace (i.e., a time series of network throughput). At each step, the agent observes the past network throughput measurement, the current video buffer size, and the remaining portion of the video. The action is the bitrate for the next video chunk. The objective is to maximize the video resolution and minimize the stall (which occurs when download time of a chunk is larger than the current buffer size) and the reward is structured to be a linear combination of selected bitrate and the stall when downloading the corresponding chunk. Prior adaptive bitrate approaches construct heuristic based on the buffer and network observations. For example, a control theoretic based approach [105] conservatively estimates the network bandwidth and use model predictive control to choose the optimal bitrate over the near-term horizon. In practice, the network condition is hard to model and estimate, making a fixed, hard-coded model-based approach insufficient to adapt to changing network conditions [5, 24, 62].

**Spark cluster job scheduling.** Efficient utilization of expensive compute clusters matters for enterprises: even small improvements in utilization can save millions of dollars at scale [12]. Cluster schedulers are key to realizing these savings. A good scheduling policy packs work tightly to reduce fragmentation [98], prioritizes jobs according to high-level metrics such as user-perceived latency [99], and avoids inefficient configurations [30]. Since hand-tuning scheduling policies is uneconomic for many organizations, there has been a surge of interest in using RL to generate highly-efficient scheduling policies automatically [22, 61, 63].

We build our scheduling system on top of the Spark cluster manager [107]. Each Spark job is represented as a DAG of computation stages, which contains identical tasks that can run in parallel. The scheduler maps executors (atomic computation units) to the stages of each job. We modify Spark's scheduler to consult an external agent at each scheduling event (i.e., each MDP step). A scheduling event occurs when (1) a stage runs out of tasks (i.e., needs no more executors), (2) a stage completes, unlocking the tasks of one or more of its children, or (3) a new job arrives in the system. At each step, the cluster has some available executors and some runnable stages from pending jobs. Thus, the scheduling agent observes (1) the number of tasks remaining in the stage, (2) the average task duration, (3) the number of executors currently working on the stage, (4) the number of available executors, and (5) whether available executors are local to the job. This set of information is embedded as features on each node of the job DAGs. The scheduling action is two-dimensional—(1) which node to work on next and (2) how many executors to assign to the node. We structure the reward at step $k$ as $r_k = -(t_k - t_{k-1})J_k$, where $J_k$ is the number of jobs in the system during the physical time interval $[t_{k-1}, t_k)$. Sum of such rewards penalize the agent in order to minimize the average job completion time. Park platform supports replaying an one-month industrial workload trace from Alibaba.

**SQL Database query optimization.** Queries in relational databases often involve retrieving data from multiple tables. The standard abstraction for combining data is through a sequential process that joins entries from two tables based on the provided filters (e.g., actor `JOIN` country `ON` actor.country_id

= country.id) at each step. The most important factor that affects the query execution time is the order of joining the tables [51]. While any ordering leads to the same final result, an efficient ordering keeps the intermediate results small, which minimizes the number of entries to read and process. Finding the optimal ordering remains an active research area, because (1) the total number of orderings is exponential in the number of filters and (2) the size of intermediate results depends on hard-to-model relationship among the filters. There have been a few attempts to learn a query optimizer using RL [51, 66, 81].

Building the sequence of joins naturally fits in the MDP formulation. At each step, the agent observes the remaining tables to join as a query graph, where each node represents a table and the edges represent the join filters. The agent then decides which edge to pick (corresponds to a particular join) as an action. Park supports rewards from a cost model (a join cost estimate provided by commercial engines) and the final physical duration. In our implementation, we use Calcite [14] as the query optimization framework, which can serve as a connector to any database management system (e.g., Postgres [83]).

**Network congestion control.** Congestion control has been a perennial problem in networking for three decades [47], and governs when hosts should transmit packets. Transmitting packets too frequently leads to congestion collapse (affecting all users) [72] while over-conservative transmission schemes under-utilize the available network bandwidth. Good congestion control algorithms achieve high throughput and low delay while competing fairly for network bandwidth with other flows in the network. Various congestion control algorithms, including learning-based approaches [27, 48, 104], optimize for different objectives in this design space. It remains an open research question to design an end-to-end congestion control scheme that can automatically adapt to high-level objectives under different network condition [89].

We implement this enviroment using CCP [74], a platform for expressing congestion control algorithms in user-space. At each step, the agent observes the network state, including the throughput and delay.[1] The action is a tuple of pacing rate and congestion window. The pacing rate controls the inter-packet send time, while the congestion window limits the total number of packets in-flight (sent but not acknowledged). We set our (configurable) action interval at 10 ms (suitable for typical Internet delays). Our reward function is adopted from the Copa [8] algorithm: `log(throughput) - log(delay)/2 - log(lost packets)`. This environment supports different network traces, from cellular networks to fixed-bandwidth links (emulated by Mahimahi [75]).

**Network active queue management.** In network routers and switches, active queue management (AQM) is a fundamental component that controls the queue size [10]. It monitors the queuing dynamics and decides to drop packets when the queue gets close to full [31]. The goal for AQM is to achieve high throughput and low delay for the packets passing through the queue. Designing a strong AQM policy that achieves this high-level objective for a wide range of network condition can be complex. Standard methods — such as PIE [42], based on PID control [9] — construct a policy for a low-level goal that maintains the queue size at a certain level. In our setting, the agent observes the queue size and network throughput measurement; it then sets the packet drop probability. The action interval is configurable (default interval 10 ms; can also go down to per packet level control). The reward can be configured as a penalty for the difference between observed and target queue size, or a weighted combination of network throughput and delay. Similar to the congestion control environment, we emulate the network dynamics using Mahimahi with a wide range of real-world network traces.

**Tensorflow device placement.** Large scale machine learning applications use distributed training environments, where neural networks are split across multiple GPUs and CPUs [69]. A key challenge for distributed training is how to split a large model across heterogeneous devices to speed up training. Determining an optimal device placement is challenging and involves intricate planning, particularly as neural networks grow in complexity and approach device memory limits [68]. Motivated by these challenges, several learning based approaches have been proposed [3, 32, 68, 69].

We build our placement system on top of Tensorflow [1]. Each model is represented as a computational graph of neural network operations. A placement scheme maps nodes to the available devices. We formulate the MDP as an iterative process of placement improvement steps [3]. At each step, the agent observes an existing placement graph and tries to improve its runtime by updating the placement

at a particular node. The state observation is the computation graph of a Tensorflow model, with features attached to each node which include (1) estimated node run time (2) output tensor size (3) current device placement (4) flag of the "current" node (5) flag if previously placed. The action places the current node on a device. Since the goal is to learn a policy that can iteratively improve placements, the reward $r_i = -(t_i - t_{i-1})$, where $t_i$ is the runtime of the placement at step $i$. Park supports optimizing placements for graphs with hundreds of nodes across a configurable number of devices. To speedup training, Park also provides a simulator for the runtime of a device placement (based on measurements from prior executions, see Appendix A4 in [3] for details).

**Circuits Design.** Analog integrated circuits often involve complex non-linear models relating the transistor sizes and the performance metrics. Common practice for optimizing analog circuits relies on expensive simulations and tedious manual tuning from human experts [85]. Prior work has applied Bayesian optimization [59] and evolution strategy [56] as general black-box parameter tuning tools to optimize the analog circuit design pipeline. [101, 102] recently proposed to use RL to end-to-end optimize the circuit performance.

Park supports transistor-level analog circuit design [85], where the circuit schematic is fixed and the agent decides the component parameters. For each schematic, the agent observes a circuit graph where each node contains the component ID, type (e.g., NMOS or PMOS) and static parameters (e.g., $Vth_0$). The corresponding action is also a graph in which each node must specify the transistor size, capacitance and resistance. Then, the underlying HSPICE circuit simulator [95] returns a configurable combination of bandwidth, power and gain as a reward. We refer the readers to [102] for more details.

**CDN memory caching.** In today's Internet, the majority of content is served by Content Delivery Networks (CDNs) [77]. CDNs enable fast content delivery by caching content in servers near the users. To reduce the content retrieval cost from a data center, CDNs aim to maximize the fraction of bytes served locally from the cache, known as the byte hit ratio (BHR) [40]. The admission control problem of CDN caching fits naturally to the MDP setting. At each step when an uncached object arrives in the CDN, the agent observes the object size, the time since the previous visit (if available) and the remaining CDN cache size. The agent then takes an action to admit or drop the uncached object. To maximize BHR, the reward at each step is the total byte hits since the last action (i.e., counting the size of cached objects served). Coupled with the admission policy is an eviction policy that decides which cached object to remove in order to make room for a newly admitted object. By default, our environment uses a fixed least-recently-used policy for object eviction. The environment also supports training an eviction agent together with the admission agent (e.g., via multi-agent RL). Our setup includes a real world trace with 500 million requests collected from a public CDN serving top-ten US websites [15].

**Multi-dim database indexing.** Many analytic queries to a database involve filter predicates (e.g., for query "SELECT COUNT(*) FROM TransactionTable WHERE state = CA AND day1 $\leq$ time $\leq$ day2", the filters are over state and time). Key to efficiently answering such range queries is the database index — the layout in which the underlying data is organized (e.g., sorted by a particular dimension). Many databases choose to index over multiple dimensions because analytics queries typically involve filters over multiple attributes [46, 106]. A good index is able to quickly return the query result by minimizing the number of points it scans. We found empirically that a well-chosen index can achieve query performance three orders of magnitude faster than one that is randomly selected. In practice, choosing a good index depends on the underlying data distribution and query workload at runtime; therefore, many current approaches rely on routine manual tuning by database administrators.

We consider the problem of selecting a multi-dimensional index from an RL perspective. We target grid-based indexes, where the agent is responsible for determining the size of the cells in the grid. We found that this type of index is competitive with traditional data structures, while offering more learnable parameters. At each step of our MDP formulation, the database receives a new set queries to run, and the agent has the opportunity to modify the grid layout. The observation consists of both the dataset (i.e., list of records in the database) and queries (i.e., a list of range boundaries for each attribute) that have arrived since the previous action. The environment then (1) samples a workload from a distribution that changes (slowly) over time, (2) uses it to evaluate the agent-generated index on a real column-oriented datastore, and (3) reports the query throughput (i.e., queries per second) as the agent's reward. Our environment uses a real dataset collected from Open Street Maps [80] with 105 million records, along with queries sampled from a set of relevant business analytic questions. In

this setup, there are more than 7 trillion possible grid layouts that the agent must encode in its action space.

**Account region assignment.** Social network websites reduce access latency by storing data on servers near their users. For each user-uploaded piece of content, the service providers must decide which region to serve the content from. These decisions have a multitude of tradeoffs: storing a piece of content in many regions incurs increased storage cost (e.g., from a cloud service provider), and storing a piece of content in the "wrong" region can substantially increase access latency, diminishing the end user's experience [6].

To faithfully simulate this effect, our environment includes a real trace of one million posts created on a medium-sized social network over eight months from eight globally distributed regions. Park supports two variants of the assignment task. First, the agent chooses a region assignment when a new piece of content is initially created. At each content creation step, the observation includes the language, outgoing links, and posting user (anonymized) ID. The action is one of the eight regions to store the content. The reward is based on the fraction of accesses from within the assigned region. This variant can be viewed as a contextual multi-armed bandit problem [57]. The second variant is similar to the first one, except that the agent has the opportunity to migrate any content to any region at the end of each 24 hour time period. The action space spans all possible mappings between the users and the regions. In this case, the agent must balance the cost of a migration against the potential decrease in access latency.

**Server load balancing.** In this simulated environment, an RL agent balances jobs over multiple heterogeneous servers to minimize the average job completion time. Jobs have a varying size that we pick from a Pareto distribution [36] with shape 1.5 and scale 100. The job arrival process is Poisson with an inter-arrival rate of 55. The number of servers and their service rates are configurable, resulting in different amounts of system load. For example, the default setting has 10 servers with processing rates ranging linearly from 0.15 to 1.05. In this setting, the load is 90%. The problem of minimizing average job completion time on servers with heterogeneous processing rates does not have a closed-form solution [39]; a widely-used heuristic is to join the shortest queue [25]. However, understanding the workload pattern can give a better policy; for example, one strategy is to dedicate some servers for small jobs to allow them finish quickly even if many large jobs arrive [29]. In this environment, upon each job arrival, the observed state is a vector $(j, s_1, s_2, ..., s_k)$, where $j$ is the incoming job size and $s_k$ is the size of queue $k$. The action $a \in \{1, 2, ..., k\}$ schedules the incoming job to a specific queue. The reward $r_i = \sum_n \left[\min(t_i, c_n) - t_{i-1}\right]$, where $t_i$ is the time at step $i$ and $c_n$ is the completion time of active job $n$.

**Switch scheduling.** Switch scheduling poses a matching problem that transfers packets from the incoming ports to the outgoing ports [60, 67, 88]. This abstracted model is ubiquitous in many real world systems, such as datacenter routers [35] and traffic junctions [44]. At each step, the scheduling agent observes a matrix of queue lengths, with element $(i, j)$ indicating the packet queue from input port $i$ to output port $j$. The matching action is bijective — no two incoming packets shall pass through the same output ports. Notice that in a switch with $n$ input/output ports, the action space is the $n!$ possible bijection matchings.[2] After each scheduling round, one packet is transferred per each input/output port pair. The goal is to maximize switch throughput while minimizing packet delay. The optimal scheduling policy for this problem is unknown and is conjectured to depend on the underlying traffic pattern [88]. For example, the max weight matching policy empirically performs well only under high load [60]. Adapting the scheduling policy under dynamics load to optimize an arbitrary combination of throughput and delay is challenging.

# B   Experiment setup

This section details the experiment setup for benchmarking existing RL algorithms in Park. We show the result of the benchmarks in Figure 4.

## B.1   RL algorithms

We follow the standard implementations of existing RL algorithms in OpenAI baselines [26]. We performed a coarse grid search for finding a good set of hyperparameters. Specifically, A2C [71]

uses separated policy and value network and it has training batch of size 64. For discrete-action environments, A2C explores using an entropy term in policy loss [71, 103], with the entropy factor linearly decay from 1 to 0.01 in 10,000 iterations. For continuous-action environments, the policy network outputs the mean of a Gaussian distribution. The variance is controlled by an external factor that decays according to the same schedule as the discrete case. In Policy Gradient (PG) [94], we rollout 16 parallel trajectory and we use a simple time-based baseline averaging the return across the trajectories. DQN [70] employs a replay memory with size 50,000 and updates the target Q network every 100 steps. DDPG [55] uses a small replay memory with 2048 objects and updates the target networks every 1000 steps.

For feed forward networks, we use simple fully connected architecture with two hidden layers of 16 and 32 neurons. For recurrent neural networks, we use LSTM with 4 hidden layers. We use graph convolution neural networks (GCNs) [50] to encode the states that involve a graph structure. In particular, we modify the message passing kernel in Spark scheduling and Tensorflow device placement problems. The kernel is $\mathbf{e}_v \leftarrow g\left[\sum_{u \in \xi(v)} f(\mathbf{e}_u)\right] + \mathbf{e}_v$, where $\mathbf{e}$ is the feature vector on each node, $f$ and $g$ are non-linear transformatio implemented by feed forward networks, $\xi(\cdot)$ denotes the child nodes. When updating the neural network parameters, we use Adam [23] as the optimizer. The non-linear activation function is Leaky-ReLU [73]. We do not observe significant performance change when changing the hyperparameter settings.

## B.2    Environment configuration and comparing baselines

**Adaptive video streaming.** We train and test the A2C agent on the simulated version of the video streaming environment since the interaction with real environment is slow. However, the learned policy can generalize to a real video environment if the underlying network conditions are similar [62]. We compare the learned A2C policy against two standard schemes. The "buffer-based" heuristic switches the bitrates purely based on the current playback buffer size [43]. "robustMPC" uses a model predictive control framework to decide the bitrate based on a combination of the current buffer size and a conservative estimate of the future network throughput [105]. We use the default parameters in the baseline algorithm from their original paper [105].

**Spark cluster job scheduling.** The benchmark experiment is on a cluster of 50 executors with a batch of 20 Spark jobs from the TPC-H dataset [96]. During training in simulation, we sample 20 jobs uniformly at random from all available jobs. We test on a real cluster with the same setup and unseen job combinations. The "fair" scheduler gives each job an equal fair share of the executors and round-robins over tasks from runnable stages to drain all branches concurrently. The "optimal weighted fair" scheduler is carefully-tuned to give each job $T_i^\alpha / \sum_i T_i^\alpha$ of the total executors, where $T_i$ is the total work of each job $i$ and $\alpha$ is a tuning factor. Notice that $\alpha = 0$ reduces to a simple fair scheme and $\alpha = 1$ reduces to a weighted fair scheme based on job size. We sweep through $\alpha \in \{-2, -1.9, ..., 2\}$ for the optimal factor.

**SQL Database query optimization.** We train and test a DQN agent on a cost model implemented in the open source query optimization framework, Calcite. This provides an estimate of the number of records that would have to be processed when we choose an edge in the query graph (apply a Join), and how long it would take to process them based on the hardware characteristics of the system. The cost model is based on the non-linear cost model ('CM2') described by [51], where the non-linearity models the random access memory constraints of a physical system. The training set, and test set, are generated from 113 queries in the Join Order Benchmark [53], with a $50\%$ train-test split. We use the following baselines from traditional database research to compare against the RL approach. *(1) Exhaustive Search:* For a given cost model, we can find the optimal policy using a dynamic programming algorithm (Exhaustive Search) and all our results are presented relative to this ($-1.00$ means the plan was as good as Exhaustive Search plan). *(2) Left Deep Search:* Is a popular baseline in practice since it finds the the optimal plan in a smaller search space (only considering join plans that form a left deep tree [51]) making it computationally much faster than Exhaustive Search.

**Network congestion control.** We train and test the A2C agent in the centralized control setting (a single TCP connection) on a simple single-hop topology. We used a 48Mbps fixed-bandwidth bottleneck link with 50ms round-trip latency and a drop-tail buffer of 400 packets (2 bandwidth-delay products of maximum size packets) in each direction. For comparison, we run TCP Vegas [18]. Vegas attempts to maintain a small number of packets (by default, around 3) in the bottleneck queue, which

results in an optimal outcome (minimal delay and packet loss, maximal throughput) for a single-hop topology without any competing traffic. "Confined search space" means we confine the action space of A2C agent to be only within $0.2$ and $2\times$ of the average action output from Vegas.

**Network active queue management.** We train and test the agent on a 10Mbps fixed-bandwidth bottleneck link with 100ms round-trip latency where there are 5 competing TCP flows. The agent examines the state and takes an action every 50ms. We configure the reward to be the current distance from the target queuing delay (20ms). As a comparison, we run "PIE" [42], a classic PID control scheme, with the same target queuing delay.

**Tensorflow device placement.** We consider device placement optimization for a neural machine translation (NMT) model [11] over two devices (GPUs). This is a popular language translation model that has an LSTM-based encoder-decoder and attention architecture to translate a source sequence to a target sequence. The training is done over a reliable simulator [3] to quickly obtain run-time estimates given a placement configuration. In the "Single GPU" heuristic, all ops are co-located on the same device, which is optimal for models that can fit in a single device and which do not have significant parallelism in their structure. Scotch [82] is a graph partitioning based heuristic that takes as input both the computational cost of each node and the communication cost along each edge. It then outputs a placement that minimizes total communication cost, while load balancing computation across the devices to within a specified tolerance. The human expert places each LSTM layer on a different device as recommended by Wu et al. [11]. PG-LSTM [69] embeds the graph model as a sequence of node features, and uses an LSTM to output the corresponding placement for each node in the sequence. The PG-GCN [3] on the other hand, uses a graph neural network [20, 38] for embedding the model, and represents the policy as performing iterative placement improvements rather than outputting a placement for all the nodes in one shot.

**Circuits Design.** The benchmark trains and tests on a fixed three-stage transimpedance amplifier analog circuit. "BO" is a simple Bayesian optimization approach to tune the model parameter. "MACE" is a prior work based on acquisition function ensemble [58]. "ES" stands for evolutional strategy approach [86]. "NG-RL" is the short of non-grach Reinforcement Learning in which we do not involve graph informantion in the optimzation loop. "GCN-RL" is the Reinforcement Learning with graph convolutional neural networks. From the results, we can observe that "GCN-RL" could consistently achieve higher Figure of Merits (FoM) value than other methods. Comparing to "NG-RL", "GCN-RL" has higher FoM value and also faster convergence speed, which indicates the critical role of the graph information.

**CDN memory caching.** We train and test A2C on several synthetic traces (10000 requests long) produced by an open-source trace generator [16]. We consider a small cache size of 1024KB for the experiment. The LRU heuristic always admits requests, with stale objects evicted based on the last recently used (LRU) policy. Offline optimal uses dynamic programming to compute the best sequence of actions, with the knowledge of future object arrivals.

**Multi-dim database indexing.** We train and test on a real in-memory column-store, using a dataset from Open Street Maps [80], comprised of 105 million points, each with 6 attributes. The dataset is unchanged across all steps. The query workload shifts continuously between different query distributions, completing a full shift to a new distribution every 20 steps. At each step, the agent observes the previous workload and produces a parametrization of the grid index that is tested on the next workload. We use a batch size of 1, and the environment is terminal at every state (i.e., the discount factor $\gamma$ is 0).

We heavily restrict the state and action space to make this environment tractable. The agent does *not* observe the underlying data, since the dataset does not change; it observes only the query workload. Each workload consists of 10 queries, each with two 6-dimensional points to specify the query rectangle, producing a 120-dimensional observation space. Each query coordinate is scaled to $[0, 1]$, relative to the range of the corresponding attribute in the OSM dataset. If an attribute is not present in the range filter, the query coordinates for that dimension are 0 and 1. For the agent's action, we fix an ordering of dimensions that we have found to work well empirically; the agent is responsible solely for determining the number of columns along each dimension in the grid, which is a 4-dimensional action space. The baseline is a fixed layout that is run on the same workloads as the agent, tuned roughly by hand to produce low running times on the *entire* sequence of workloads. The baseline layout uses the same dimension ordering that was fixed for the agent and is not re-optimized for each new workload.

**Account region assignment.** The setup for this experiment follows the first variant of the assignment task outlined in Appendix A, in which the agent has to assign newly created accounts to one of eight regions. Local heuristic is a simple baseline that assigns an account directly to the region it was created in. The Thompson sampling [21] approach uses a random forest model comprising of 100 trees. We train and test DQN over the real trace of one million posts included with Park.

**Server load balancing.** In this experiment we consider the setup as described in Appendix A, with 10 heterogenous servers. The A2C [71] learning approach is elaborated in Appendix B.1; 'grad clip' refers to gradient clipping, in which we normalize the policy gradient by its l2 norm when the l2 norm is over 10. The greedy heuristic assigns each incoming job to that queue having the lowest queue size to processing rate ratio.

**Switch scheduling.** We consider scheduling in a crossbar switch (Appendix A) with 3 input ports and 3 output ports. Time is discretized for simplicity. Traffic between each port pair $(i, j)$ is generated according to a Bernoulli process, with rate given by the $(i, j)$-th entry of a random bistochastic traffic matrix. The load of the system (i.e., the row and column sums of the traffic matrix) is set to 90%. MWM, or Max-Weight-Matching [88], is a well-known scheduling policy that forwards packets at each time-step according to the maximum weighted matching on the bipartite graph between the set of input and output ports. The weight of each edge $(i, j)$ on the bipartite graph is set equal to the size of the virtual-output queue (VOQ) $j$ at input port $i$ [88]. For a parameter $\alpha > 0$, MWM-$\alpha$ refers to an analogous policy where the weight of edge $(i, j)$ on the bipartite graph is set equal to the size of VOQ $j$ at input port $i$ raised to the power $\alpha$.

## Footnotes

[1]See Table 2 of [74] for full list.

[2]Typical routers can have 144 ports [37].