[Reviews · NeurIPS 2019]

Reviewer 1



4) Originality: The work is highly original. It introduces a suite of real-world environments where RL algorithms can have an immediate impact. 5) Quality: This submission is of high quality. The environments are well-described and the experiments are exhaustive. The authors also highlight the limitations of their environment e.g. the size of the cache. 6) Clarity: The work is well-written. It makes a good job of describing the particular challenges of using RL algorithms for computer systems. 7) Significance: I think the work is significant and I hope it encourages other researchers to develop RL methods designed for computer systems. 8) Conclusion: I recommend acceptance given the importance of real-world environments to benchmark RL algorithms.

Reviewer 2



It is great to see the kind of interest in applying machine learning, and specifically reinforcement learning, into real-world problems such as computer systems as presented in this paper. While the paper has no significant contributions on either a theoretical or algorithmic front, it does an important job at highlighting some of the issues in applying modern RL algorithms to real problems, and provides a necessary benchmarking environment for computer systems research specifically. The problem domains included have a wide variety of characteristics, from high-frequent real-time systems to very-long horizon problems, uniquely structured state and action spaces and both simulated and real environments (some other related work that could be added is [1]). Especially the latter is valuable to ground any research. Moreover, the authors provide an RL baseline result for each of the proposed tasks, and highlight some of the problematic characteristics of these tasks for RL specifically. There could be a more elaborate discussion of the results however. Overall the paper is clearly written and structured, although there are some minor grammatical errors (mainly in the appendices). [1] Claeys, Maxim, et al. "Design and evaluation of a self-learning HTTP adaptive video streaming client." IEEE communications letters 18.4 (2014): 716-719. UPDATE I thank the authors for their rebuttal and agreeing to incorporate my suggestions. I am sticking with my score.

Reviewer 3



This paper (alongwith its supplementary material) provides good detail of the Park platform and the hyperparameter settings during the experiments which makes the work reproducible. The paper is well written and the flow is understandable too. The paper claims Park to be an extensible platform at more than one occasions, however, the claim has not been substantiated with any metric. What makes the platform extensible should be more elaborate in the paper. For example may be a case study can help elaborate how easy is it to add a new systems environment? How much effort is required? etc. The paper should discuss how different/good/bad is Park from similar or the closest platforms s.a. Facebook Horizon. What makes Park different from others and how? If Park has outperformed any baselines in other environments/problem specific learning methods, those comparisons can be made part of the paper to show superiority of Park There are a few horizontal lines (i.e. no improvement in policies is seen) in figure 4. These cases should be discussed in text why has this happened? any possible reasons for such situations Line 284: The paper should further elaborate which existing heuristics and optimal policies have been provided (just name them in the paper as an example for ease of the reader)

[Author Response · NeurIPS 2019]

We thank the reviewers for their interest in the paper and their constructive feedback. In the following, we address the
outstanding comments and outline our plan for improving the paper.

**Comparison to Facebook's Horizon platform (Reviewer 3).** Face-
book's Horizon platform plays an orthogonal and complementary role
to Park. In Figure 1, we show where Horizon and Park reside in a
typical RL workflow diagram. As shown, Horizon implements several
RL algorithms and exposes APIs to developers to integrate RL training
and maintenance into their systems. By contrast, Park is a research
platform that provides a common interface to connect to a wide range
of computer system environments. In fact, Park's primary contribution
is this environment suite that can serve as benchmarks for RL research
in computer systems. In principle, a developer can use Horizon to
try out existing RL algorithms or develop new RL algorithms for the
system environments in Park. We will add this comparison to Horizon
to the introduction (where we compare with OpenAI Gym) to better position the Park platform.

**Figure 1:** RL workflow. Park uses a common interface to connect to twelve computer system environments.

**"Correctness" of system MDP (Reviewer 3).** Thank you for raising this point. Formulating the MDP is an important,
problem-specific step in applying RL to any system. In fact, some of Park's systems have multiple possible MDP
formulations, which can affect the efficiency of RL training (e.g., for the Tensorflow device placement problem, an
"incremental improvement" formulation (Addanki et al., 2019) is more efficient than the "one-shot" MDP (Mirhoseini
et al., 2018)). While the main contribution of Park is not defining the MDP for each system environment, we thought
carefully to pick an appropriate MDP formulation for each problem. Our guiding principle was to provide the RL agent
with all the information and actions available to existing baselines schemes in that environment, such that the agent could
at least express existing human-engineered policies. In most cases, the MDP formulations are straightforward and their
correctness is self-explanatory. However, some are more subtle (e.g., the Spark scheduling and TF device placement),
and in these cases we adopt the formulations from prior work. In the final version of the paper, we will discuss the MDP
formulation for each system in more detail in the appendix, pointing out any subtleties with reference to prior work.

**How extensible is Park's interface? (Reviewer 3).** We will use lines of code added for each system to quantify how
much work it takes to add new environments with Park's interface. For example, the adaptive video streaming environment
uses 324 additional lines of code for Park interaction. To add a new environment, as we commented in §4, the developer
only needs to specify the state and action space, and make them accessible through RPCs. We will add the additional lines
of code for each environment as a reference in the appendix.

**Details of baseline schemes (Reviewer 3).** In our experiment, we compare against existing heuristics specifically
designed for each environment. In the appendix, we describe the details for the compared heuristics, the overview of their
implementations and the intuitions for how they work. Due to space limit, we only provide the names of these baseline
schemes in Figure 4. We will add a forward pointer to the appendix in the experiment section for the details.

**More detailed discussions on the experimental results (Reviewer 2 and 3).** We agree with Reviewer 2 about trimming
§4 (mostly the "Real System Interaction Loop" part) and adding more details from the appendix to §5. As suggested,
we will highlight and comment on the results where the RL policies are not stable. We will also describe our model
tuning strategy, a form of grid search, which is performed separately for each environment. We believe that the instability
observed in some of the environments are due to the fundamental challenges discussed in the paper, not poorly tuned
training procedure. For example, the policy in Figure 4(h) is unable to converge smoothly partially due to the variance
caused by the cache arrival input sequence (i.e., the challenge in §3.2). Also, to clarify, the horizontal lines in Figure 4
correspond to fixed, non-learning baseline schemes in each system. These policies are not trained. We include their
performance level as a constant line for reference in Figure 4.

**Standardized interface for testing environments (Reviewer 1).** In our experiments, we test the RL agent on unseen
scenarios by loading a different set of traces. For example, the adaptive video streaming environment loads unseen
network traces and the Tensorflow device placement environment tests with unseen models (i.e., unseen computation
graphs). The users have the flexibility to create different scenarios to test the robustness of a learning model. However, we
agree that standardizing the training/testing scenarios (e.g., fixing which traces to use for training/testing) would provide
a better common ground for comparison. We will provide flags to launch default training and testing scenarios in our
implementation.

**Elaborate on future directions (Reviewer 1 and 3).** In the conclusion section, we will highlight the future research
direction by summarizing the challenges discussed in §3. In addition, the rightmost column of Table 2 outlined the
challenges for each environment. We will make them stand out more when we discuss the table in §4.

**Adding reference (Reviewer 2).** We will add the citation for adaptive video streaming. Thanks for pointing this out!

[Meta-Review · NeurIPS 2019]

The reviewers have each reviewed this paper carefully, and have taken the author response into account. There is clear consensus among them that this paper is a valuable contribution to the research community, both in helping to bring the application area of ML for systems environment more into the conversation and for providing a solid suite of benchmarks to foster further innovation within the community. I especially appreciate this aspect of helping to make the future research community more effective. In the author response, the authors describe several ways in which their paper will be revised to take reviewer feedback into account, and I expect this will be done for any final version of the paper.